# Personalized Federated Recommendation with Knowledge Guidance

## Abstract

Federated Recommendation (FedRec) has emerged as a key paradigm for building privacy-preserving recommender systems. However, existing FedRec models face a critical dilemma: memory-efficient single-knowledge models suffer from a suboptimal knowledge replacement practice that discards valuable personalization, while high-performance dual-knowledge models are often too memory-intensive for practical on-device deployment. We propose Federated Recommendation with Knowledge Guidance (FedRKG), a model-agnostic framework that resolves this dilemma. The core principle, Knowledge Guidance, avoids full replacement and instead fuses global knowledge into preserved local embeddings, attaining the personalization benefits of dual-knowledge within a single-knowledge memory footprint. Furthermore, we introduce Adaptive Guidance, a fine-grained mechanism that dynamically modulates the intensity of this guidance for each user-item interaction, overcoming the limitations of static fusion methods. Extensive experiments on benchmark datasets demonstrate that FedRKG significantly outperforms state-of-the-art methods, validating the effectiveness of our approach.

## 1 Introduction

Recommendation systems are integral to modern online services, but their reliance on sensitive user data raises significant privacy concerns under regulations like GDPR (Voigt & Von dem Bussche, 2017). Federated Learning (FL) (McMahan et al., 2017) offers a compelling solution by enabling decentralized model training without sharing raw data. This has led to a surge in research on Federated Recommendation (FedRec), which aims to deliver personalized experiences while preserving user privacy (Zhang et al., 2024; Li et al., 2024; 2025).

Existing FedRec models fall into two categories: single-knowledge and dual-knowledge approaches. Single-knowledge models require clients to store one set of item embeddings. However, they follow a suboptimal *knowledge replacement* practice, where locally personalized embeddings are completely overwritten by the global model in each round, discarding valuable specialized information (Ammad-Ud-Din et al., 2019; Chai et al., 2020; Perifanis & Efraimidis, 2022; Zhang et al., 2023; 2024; He et al., 2024). In contrast, dual-knowledge models (Li et al., 2024; 2025) achieve superior personalization by maintaining separate global and local knowledge components. This performance gain, however, comes at the cost of *doubling the memory footprint*, making them impractical for resource-constrained on-device environments like mobile phones where FedRec is most needed.

This dilemma—choosing between the memory efficiency of single-knowledge models and the personalization of dual-knowledge ones—highlights a critical trade-off. We challenge the necessity of the knowledge replacement practice in single-knowledge models. Our key experimental finding reveals that periodically injecting global knowledge into a local model, rather than replacing local knowledge, causes a step-wise surge in performance. This suggests that fusing global trends with preserved local specializations is a more effective approach. Based on this insight, we introduce **Knowledge Guidance**, a simple yet effective learning paradigm that attains dual-knowledge-level performance while keeping a single set of embeddings, thereby avoiding memory overhead. Importantly, we also provide a theoretical interpretation: the guidance is equivalent to a global knowledge regularized update (i.e., one-step gradient descent on a quadratic global knowledge alignment penalty) rather than an ad-hoc heuristic. A conceptual comparison is shown in Figure 1.

Figure 1: A conceptual comparison between (a) Single Knowledge, (b) Dual Knowledge, and (c) Guided Knowledge (Ours)

Building upon the core concept of Knowledge Guidance, we further refine how this guidance is applied. Prior dual knowledge models apply global knowledge with a static intensity, either uniformly across all items (Li et al., 2024) or fixed per user (Li et al., 2025). We argue this is suboptimal, as the ideal balance between personalization and generalization varies for each user-item interaction. We propose **Adaptive Guidance**, a fine-grained guidance mechanism that dynamically modulates the intensity of global knowledge for each interaction, supported by a novel nested training process. To this end, we propose **Fed**erated **R**ecommendation with **K**nowledge **G**uidance (**FedRKG**), a unified, model-agnostic framework with Guidance mechanism.

Our contributions are as follows:

- We identify and address the critical trade-off between memory cost and personalization in FedRec, challenging the suboptimal knowledge replacement practice in single-knowledge models and doubling memory in dual-knowledge models.

- We introduce **Knowledge Guidance**, a novel paradigm that transitions FedRec from the conventional *replacement* of knowledge to a more effective *guidance* model. This memory-efficiently fuses global trends and local specialization within a single set of embeddings.

- We further extend Knowledge Guidance to **Adaptive Guidance**, a mechanism that dynamically learns the optimal intensity of global guidance for each user-item interaction, enabled by a novel nested training process.

- Our framework is model-agnostic, offering broad applicability to existing single-knowledge recommendation models, and extensive experiments demonstrate that FedRKG substantially outperforms state-of-the-art competitors on real world benchmark datasets.

## 2 RELATED WORK

**Personalized Federated Learning.** Federated Learning (FL) enables the training of a model with performance comparable to traditional centralized methods by aggregating models trained on local client data, without collecting the data (McMahan et al., 2017). However, this approach, which assumes that all clients have an identical data distribution, fails to consider the data heterogeneity across clients. To address this issue, Personalized Federated Learning (PFL) has emerged. PFL aims to build personalized models for each client(Tan et al., 2022). Several lines of research have explored different PFL strategies. Some studies introduce regularization terms between the local and global models (Ren et al., 2024; Long et al., 2024; Tang et al., 2024; Zhang et al., 2024). Another approach involves model mixup-based PFL, where models are divided into shared and personalized components. Some works use heterogeneous encoders to extract personalized knowledge while using a homogeneous decoder for interpretation (Jang et al., 2022; Liu et al., 2022; Yi et al., 2023). Other studies implement the opposite structure, utilizing a shared encoder with local decoders or classifiers(Pillutla et al., 2022; Collins et al., 2021; Oh et al., 2021). There are also efforts that leverage meta-learning and hypernetworks to achieve personalization (Scott et al., 2023; Lim et al., 2024).

**Federated Recommendation.** Federated Recommendation applies FL to recommender systems to protect sensitive user data (Ammad-Ud-Din et al., 2019; Sun et al., 2024; Li et al., 2024; 2025).

Early works adapts centralized models, such as Matrix Factorization (Koren et al., 2009) and Neural Collaborative Filtering (He et al., 2017), to FL settings (Ammad-Ud-Din et al., 2019; Chai et al., 2020; Perifanis & Efraimidis, 2022). Later studies explore advanced techniques including privacy-aware Graph Neural Networks (Wu et al., 2022), while others construct a graph from user updates to create personalized item embeddings (Zhang et al., 2024). Other efforts include developing personalized score functions and two-step optimization (Zhang et al., 2023), introducing self-supervised pre-training (Luo et al., 2024), utilizing group-wise information (He et al., 2024). Most follow a single-knowledge paradigm. In contrast, recent dual-knowledge models show superior performance by separating global and local representations. FedRAP (Li et al., 2024) maintains local and global item embeddings per client, while FedDAE (Li et al., 2025) uses a VAE-based architecture (Kingma et al., 2013) with separate encoders and a gating network to balance their influence.

**Remarks.** Distinct from these approaches, our work introduces a novel paradigm within the single-knowledge framework. Instead of completely replacing local embeddings, we propose Knowledge Guidance that preserves personalized knowledge while periodically injecting global trends. This allows our model to operate with the memory efficiency of a single-knowledge system while emulating the high personalization performance characteristic of dual-knowledge approaches.

## 3 PROBLEM FORMULATION

Let $\mathcal{U}$ and $\mathcal{I}$ be the sets of $n$ users and $m$ items, respectively. User-item interactions are represented by a matrix $\mathbf{R} \in \{0,1\}^{n \times m}$, where $r_{ui} = 1$ indicates that user $u \in \mathcal{U}$ has interacted with item $i \in \mathcal{I}$, and $r_{ui} = 0$ otherwise. For each user $u$, we define the set of interacted items as $\mathcal{I}_u^+ = \{i \in \mathcal{I} \mid r_{ui} = 1\}$ and unobserved items as $\mathcal{I}_u^- = \{i \in \mathcal{I} \mid r_{ui} = 0\}$. Each user $u$ maintains local parameters $\theta_u = \{\mathbf{P}_u, \mathbf{e}_u, \mathbf{M}_u\}$, consisting of a personalized item embedding matrix $\mathbf{P}_u \in \mathbb{R}^{m \times d}$, a private user embedding vector $\mathbf{e}_u \in \mathbb{R}^d$, and, if necessary, a user-specific scoring function $\mathbf{M}_u$. In our framework, the user embedding and scoring function remain strictly on the client and are never shared with the server, ensuring that direct user preferences are protected. A client-side recommendation model $\mathcal{F}_u$ predicts user $u$'s preference for item $i$ as $\hat{r}_{ui} = \mathcal{F}_u(i; \theta_u)$, where $\theta_u$ denotes the local parameters (e.g., $\hat{r}_{ui} = \sigma(\mathbf{M}_u(\mathbf{e}_u, \mathbf{P}_{ui}))$). The model then generates a personalized recommendation list by ranking scores of all unobserved items $i \in \mathcal{I}_u^-$. We train $\mathcal{F}$ under a federated setting. Each client corresponds to a single user. In each communication round $t$, a subset of users $\mathcal{U}^t \subseteq \mathcal{U}$ is sampled. In standard rounds, clients in $\mathcal{U}^t$ minimize recommendation loss $\mathcal{L}_{rec}$ (Eq.2) on their local devices without parameter replacement, whereas in guidance rounds they additionally apply Knowledge Guidance.

**Federated Optimization Objective.** We formulate the task as a personalized federated learning problem that minimizes the sum of local objective functions:

$$\min_{\{\theta_1,\ldots,\theta_n\}} \sum_{u \in \mathcal{U}} \mathcal{L}_{rec}(\theta_u), \quad (1)$$

where $\mathcal{L}_{rec}(\theta_u)$ is the local objective function for user $u$.

**Recommendation Model Loss.** We consider a standard recommendation setting based on implicit feedback, where user-item interactions are recorded as binary values. The goal is to learn the model parameters $\theta_u$ by maximizing the likelihood of observing positive interactions over negative ones. To this end, we define the recommendation loss as the binary cross-entropy loss, formulated as:

$$\mathcal{L}_{rec}(\theta_u) = -\sum_{i \in \mathcal{I}_u^+} \log \hat{r}_{ui} - \sum_{i \in \mathcal{D}_u^-} \log(1 - \hat{r}_{ui}), \quad (2)$$

where $\mathcal{D}_u^- \subset \mathcal{I}_u^-$ is a sampled set of negative items.

## 4 PROPOSED FRAMEWORK: FEDRKG

In this section, we present **Fed**erated **R**ecommendation with **K**nowledge **G**uidance (**FedRKG**). Our approach is grounded in two design principles: (i) preserve personalized item embeddings and use global items embeddings only as guidance to complement them; and (ii) apply this guidance adaptively, allowing its intensity to vary across users and items. We first propose a simple yet powerful

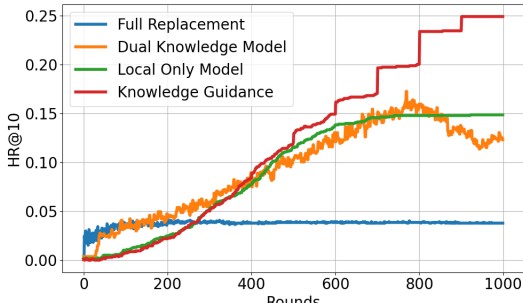

Figure 2: Convergence on Amazon-Video (HR@10 vs. rounds) with a FedMF backbone. Methods: Full Replacement (single-knowledge), Local Only, Dual-Knowledge (FedRAP), and Knowledge Guidance (ours). Guidance every 100 rounds yields stepwise improvements and the best final HR@10 while retaining a single-embedding inference footprint.

training principle, Knowledge Guidance, from an empirical observation. Building on this principle, we introduce FedRKG, a practical federated recommendation framework that operationalizes Knowledge Guidance as Adaptive Guidance. Finally, we describe the nested training required to learn the adaptive gate. The overall architecture is shown in Figure 3; the overall flow is detailed in Algorithm 1 in the Appendix.

## 4.1 FROM OBSERVATION TO THE KNOWLEDGE GUIDANCE PRINCIPLE

We conduct experiments on Amazon-Video (Ni et al., 2019) dataset using FedMF (Chai et al., 2020) as the backbone and set the local epoch to one per round to closely track per-round behavior. Figure 2 highlights a core pathology of prior paradigms. Full Replacement yields the worst performance, indicating that repeatedly discarding personalization is harmful. Local-Only fares better by preserving local knowledge but stagnates without global trends. In contrast, our Knowledge Guidance curve exhibits distinct step-wise gains exactly when guidance is applied, indicating that periodically fusing the global model into preserved local embeddings rather than replacing them synergistically combines generalization and personalization (formalized in Section 4.2 via the update in Eq. 3). Crucially, this approach outperforms a heavier dual-knowledge baseline (e.g., FedRAP (Li et al., 2024)), yielding a superior utility–memory trade-off. These observations motivate the guidance principle (Knowledge Guidance)—periodically fusing global knowledge into preserved local embeddings (Section 4.2). Building on this principle, we develop **FedRKG**, a practical framework that instantiates and extends Knowledge Guidance with Adaptive Guidance for interaction-specific modulation (Section 4.3).

## 4.2 KNOWLEDGE GUIDANCE

At pre-defined synchronization points (e.g., every $T_{int}$ rounds with $T_{int} \geq 1$), the server aggregates the item embeddings from all participating users to form the global item embeddings $\mathbf{P}_g$ (e.g., via FedAvg; $\mathbf{P}_g = 1/n \sum_{u \in \mathcal{U}} \mathbf{P}_u$), which are then distributed back to the clients. Each client $u$ updates personalized embeddings $\mathbf{P}_u$ with $\mathbf{P}_g$:

$$\mathbf{P}_u \leftarrow \beta\,\mathbf{P}_u + (1-\beta)\,\mathbf{P}_g, \tag{3}$$

where $\beta \in [0,1]$ controls how strongly personalized knowledge is preserved rather than assimilated into global trends. This yields the benefits of dual-knowledge training while retaining single-knowledge memory at inference.

**Theoretical interpretation.** **Proposition 1.** Updating Eq. 3 is equivalent to one gradient-descent step on $\mathcal{L}(\mathbf{P}) = \frac{\lambda}{2}\|\mathbf{P} - \mathbf{P}_g\|_2^2$ with step size $\eta = \frac{1-\beta}{\lambda}$ ($\lambda > 0$). *Proof* is provided in the Appendix. We prove that Knowledge Guidance is a global knowledge regularized update that gently pulls $\mathbf{P}$ toward the global knowledge $\mathbf{P}_g$ while preserving local specialization.

Figure 3: Overall framework of FedRKG. On each client $u$: (i) **Before Guidance**, the local model is trained with $\mathcal{L}_{rec}$ (Eq.2); (ii) **Nested Training**, a lightweight gate is fit (main parameters frozen) to produce an adaptive guidance vector; (iii) **During Guidance**, global knowledge is injected into the single local embedding set using guidance vector (applied twice here), avoiding full replacement and preserving personalization.

## 4.3 FEDRKG: ADAPTIVE GUIDANCE

While naive guidance is effective, using a single scalar ($\beta$) for all items is suboptimal: niche items often require stronger personalization, whereas popular items can lean more on global knowledge. FedRKG therefore learns interaction-specific guidance via an adaptive gating mechanism.

### 4.3.1 ADAPTIVE GATING MECHANISM

For each user–item pair $(u, i)$, we form a guidance vector by scaling the global item embedding $\mathbf{P}_{gi}$ with a learned gate $g_{ui} \in (0, 1)$:

$$\mathbf{G}_{ui} = 2\, g_{ui}\, \mathbf{P}_{gi}. \tag{4}$$

The gate is computed by a user-specific gating network that takes the relationship between the personalized and global embeddings as input:

$$g_{ui} = \sigma\big(\mathbf{W}_u^\top \big[\mathbf{P}_{ui} : \mathbf{P}_{gi} : \mathbf{P}_{ui} - \mathbf{P}_{gi}\big] + b_u\big), \tag{5}$$

where $[\cdot : \cdot]$ denotes concatenation and $\sigma(\cdot)$ is the sigmoid.

$$\mathbf{P}_{ui} \leftarrow \beta\, \mathbf{P}_{ui} + (1 - \beta)\, \mathbf{G}_{ui} = \beta\, \mathbf{P}_{ui} + 2\,(1 - \beta)\, g_{ui}\, \mathbf{P}_{gi}. \tag{6}$$

### 4.3.2 NESTED TRAINING FOR THE GATE

A difficulty is that guidance is an update rule, not a standard differentiable layer. FedRKG addresses this with a two-step nested procedure at each guidance round:

**Step 1: Local gate training.** During the client-side gate training for client $u$, the local optimizer updates only the gating parameters $\{\mathbf{W}_u, b_u\}$; all other recommender parameters $\{\mathbf{P}_u, e_u, \mathbf{M}_u\}$ and the server-provided global item embeddings $\mathbf{P}_g$ are frozen. We replace the standard item embeddings with the fused temporary embeddings from Eq. 6 (combining $\mathbf{P}_u^t$, $\mathbf{P}g^t$, and the gate), and minimize the recommendation objective on local data:

$$\mathcal{L}_{rec}(\mathbf{W}_u, b_u; \mathbf{P}_g, \mathbf{P}_u, \mathbf{e}_u, \mathbf{M}_u), \tag{7}$$

where only the gating parameters receive gradients, with all other parameters are frozen. For notational simplicity, this objective can be written as $\mathcal{L}_{rec}(\mathbf{W}_u, b_u)$, and the gating parameters are updated as:

$$\mathbf{W}_u^* \leftarrow \mathbf{W}_u - \eta_{gate} \nabla_{\mathbf{W}_u} \mathcal{L}_{rec}(\mathbf{W}_u, b_u), \quad b_u^* \leftarrow b_u - \eta_{gate} \nabla_{b_u} \mathcal{L}_{rec}(\mathbf{W}_u, b_u). \tag{8}$$

Thus, we train the gating network to capture user–item–specific behavior so that the resulting guidance improves recommendation performance.

**Step 2: Apply definitive guidance.** Using the updated gating parameters $\mathbf{W}_u^*, b_u^*$ together with $\mathbf{P}_u$ and $\mathbf{P}_g$, we recompute $g_{ui}^*$ via Eq. 5 and obtain the corresponding guidance vector $\mathbf{G}_{ui}^*$ using Eq. 4. This refined guidance vector is then used to commit the fusion update:

$$\mathbf{P}_{ui} \leftarrow \beta\,\mathbf{P}_{ui} + (1-\beta)\,\mathbf{G}_{ui}^*. \tag{9}$$

After this step, training resumes on the main local model until the next guidance round.

## 5 DISCUSSION

### 5.1 MEMORY ANALYSIS

We analyze the memory space of our method, with FedMF (Chai et al., 2020) as the backbone. The client's memory footprint fluctuates depending on the training phase. The local model for each client $u$ consists of $\{\mathbf{P}_u, \mathbf{e}_u \mathbf{W}_u, b_u\}$. During most training stages, the client only needs to store its own personalized parameters. The total number of parameters is $md + 4d + 1$. During guidance, the client temporarily downloads $\mathbf{P}_g$ from the server. This increases the peak memory requirement to $2md + 4d + 1$. For inference, the knowledge from the global embedding is already fused into the client's $\mathbf{P}_u$. Therefore, the client only stores personalized parameters, requiring $md + 4d + 1$. The server's role is to aggregate parameters from clients. During a guidance stage, the server temporarily receives all $\{\mathbf{P}_u | u \in \mathcal{U}\}$, leading to a peak memory usage of $nmd$. However, after the aggregation process, the server only needs to store $\mathbf{P}_g$, which stores $md$.

Crucially, unlike dual-knowledge models that consistently demand doubling memory throughout all stages, FedRKG achieves strong performance with a significantly lower typical and inference-time memory footprint. This efficiency makes our model a more practical and scalable solution for resource-constrained devices.

### 5.2 LOCAL DIFFERENTIAL PRIVACY

In the FedRKG framework, clients transmit their item embeddings $\mathbf{P}_u$ during guidance steps, which poses a potential risk of information leakage. A malicious server can infer sensitive information by differencing a client's embeddings across two consecutive guidance rounds $r$ and $r+T_{\text{int}}$, i.e., comparing $\mathbf{P}_u^r$ and $\mathbf{P}_u^{r+T_{\text{int}}}$. Let $\Delta\mathbf{P}_u \triangleq \mathbf{P}_u^{r+T_{\text{int}}} - \mathbf{P}_u^r$ denote the accumulated local update over this interval. With learning rate $\eta$, summing the per-step updates over $T_{\text{int}}$ rounds with $E$ local epochs per round gives $\Delta\mathbf{P}_u \triangleq \mathbf{P}_u^{r+T_{\text{int}}} - \mathbf{P}_u^r \approx -\eta \sum_{t=0}^{T_{\text{int}}-1} \sum_{e=0}^{E-1} \nabla\mathcal{L}(\mathbf{P}_u^{r+t,e})$ where $\nabla\mathcal{L}(\cdot)$ denotes the gradient with respect to $\mathbf{P}_u$ (i.e., $\nabla\mathcal{L}(\cdot) \equiv \nabla_{\mathbf{P}_u}\mathcal{L}(\cdot)$), and we omit the subscript for brevity. Because these local gradients are driven by the user's private interactions, such parameter differences can reveal sensitive information (Chai et al., 2020; Li et al., 2024).

To mitigate this risk, we employ an $(\epsilon, \delta)$-Local Differential Privacy (LDP) to protect the client's information during the local training $\mathbf{P}_u$ (Minto et al., 2021; Li et al., 2024). A randomized mechanism $\mathcal{M}$ satisfies $(\epsilon, \delta)$-LDP if for any two adjacent datasets $\mathcal{D}$ and $\mathcal{D}'$ and for any set of possible outputs $\mathcal{S}$, it holds that $\Pr[\mathcal{M}(\mathcal{D}) \in \mathcal{S}] \le e^\epsilon \Pr[\mathcal{M}(\mathcal{D}') \in \mathcal{S}] + \delta$. In our setting, adjacency is user-level: two global datasets differ only in the local dataset of a single client $u^\star$; the mechanism outputs the sanitized accumulated update, and $\mathcal{S}$ ranges over measurable subsets of this output space.

Our LDP mechanism is a two-step process. First, during local training, we clip the L2-norm of each gradient to a predefined threshold $C : \nabla\mathcal{L}(\mathbf{P}_u) \leftarrow \nabla\mathcal{L}(\mathbf{P}_u) \cdot \min(1, C/\|\nabla\mathcal{L}(\mathbf{P}_u)\|_2)$. Bounding the norm of each gradient allows us to determine the sensitivity of the total parameter update between two guidance rounds. The sensitivity is defined as the maximum possible L2-norm of the difference between the final parameters, $\mathbf{P}_u^{r+T_{int}}$ and $\mathbf{P}_u'^{r+T_{int}}$, derived from two adjacent training scenarios. Assuming the parameters are identical at the beginning of the interval ($\mathbf{P}_u^r = \mathbf{P}_u'^r$), the sensitivity can be bounded as follows:

$$\max_{\mathbf{P},\mathbf{P}'} \|\mathbf{P}_u^{r+T\text{int}} - \mathbf{P}u'^{r+T\text{int}}\|_2 = \max_{\mathbf{P},\mathbf{P}'} \|\Delta\mathbf{P}_u - \Delta\mathbf{P}_u'\|_2 \tag{10}$$

$$\le \max_{\mathbf{P},\mathbf{P}'} \left( \|\Delta\mathbf{P}_u\|_2 + \|\Delta\mathbf{P}_u'\|_2 \right) \tag{11}$$

The norm of the total update for a single training path $||\Delta\mathbf{P}_u||_2$, can be bounded using the triangle inequality and the gradient clipping threshold $C$:

$$||\Delta\mathbf{P}_u||_2 = \left\|-\eta \sum_{t=0}^{T_{int}-1} \sum_{e=1}^{E} \nabla\mathbf{P}_u^{r+t,e}\right\|_2 \tag{12}$$

$$\leq \eta \sum_{t=0}^{T_{int-1}} \sum_{e=1}^{E} ||\nabla\mathbf{P}_u^{r+t,e}||_2 \tag{13}$$

$$\leq \eta T_{int} EC \tag{14}$$

Therefore, the sensitivity, which represents the maximum possible difference, is bounded by $2\eta T_{\text{int}}EC$. With this defined sensitivity, we can add noise drawn from Gaussian distribution $\mathcal{N}(0, \sigma^2)$ to the gradients of $\mathbf{P}_u$, where $\sigma$ is proportional sensitivity $C$. Following (Abadi et al., 2016; Li et al., 2024), we leverage the moment accountant to determine the final privacy budget $\epsilon$ for a given $\delta$, thereby satisfying $(\epsilon, \delta)$-LDP for the client's transmitted information.

# 6 EXPERIMENTS

## 6.1 EXPERIMENTAL SETTINGS

We introduce experimental settings in this section. Further details are provided in the Appendix.

**Datasets.** We evaluate FedRKG on four widely used benchmarks: Amazon-Video (Ni et al., 2019), FilmTrust (Guo et al., 2013), LastFM-2K (Cantador et al., 2011), and ML-1M (Harper & Konstan, 2015). Preprocessing follows previous work (Li et al., 2025).

**Evaluation.** We evaluate performance using two standard ranking metrics: Recall (R@K) and Normalized Discounted Cumulative Gain (N@K)(He et al., 2015), with K set to 5 and 10. All results are reported as percentages. To ensure a fairer comparison, unlike prior studies(Li et al., 2024; Zhang et al., 2023; 2024; He et al., 2024), we rank against all unobserved items when measuring performance following (Li et al., 2025).

**Baselines.** We compare the performance of FedRKG against several state-of-the-art methods from both centralized and federated settings. Our centralized baselines include **MF** (Koren et al., 2009), **NeuMF** (He et al., 2017). Single-knowledge federated baselines include **FedMF** (Chai et al., 2020), **FedNCF** (Perifanis & Efraimidis, 2022), **PFedRec** (Zhang et al., 2023), **GPFedRec** (Zhang et al., 2024), **CoFedRec** (He et al., 2024). We include two pioneering dual-knowledge models: **FedRAP** (Li et al., 2024), **FedDAE** (Li et al., 2025).

| | Model | CenRec | | FedRec w/ S.K | | | | | FedRec w/ D.K | | Ours | Improv |
|---|---|---|---|---|---|---|---|---|---|---|---|---|
| | | MF | NeuMF | FedMF | FedNCF | PFedRec | GPFedRec | CoFedRec | FedRAP | FedDAE | | |
| Amazon-Video | N@5 | 2.67 | 2.20 | 3.37 | 0.72 | 4.26 | 1.34 | 3.09 | 7.53 | 2.18 | **15.74** | 109.0% |
| | R@5 | 3.88 | 3.50 | 4.84 | 1.25 | 5.47 | 2.14 | 4.48 | 10.54 | 3.42 | **20.63** | 95.73% |
| | N@10 | 3.35 | 2.78 | 3.93 | 1.23 | 4.61 | 1.89 | 3.53 | 8.82 | 2.77 | **17.50** | 98.41% |
| | R@10 | 5.99 | 5.31 | 6.59 | 2.85 | 6.58 | 3.56 | 5.85 | 14.58 | 5.24 | **26.08** | 78.88% |
| FilmTrust | N@5 | 26.43 | 28.64 | 19.28 | 18.57 | 33.85 | 27.04 | 26.75 | 92.82 | 19.70 | **99.20** | 6.87% |
| | R@5 | 31.31 | 33.94 | 23.16 | 21.07 | 35.52 | 32.94 | 33.92 | 95.96 | 27.63 | **99.82** | 4.02% |
| | N@10 | 28.71 | 30.85 | 21.78 | 21.08 | 35.23 | 29.62 | 29.23 | 92.99 | 22.22 | **99.26** | 6.32% |
| | R@10 | 38.40 | 40.78 | 31.05 | 28.96 | 41.03 | 40.96 | 41.52 | 97.39 | 35.56 | **100.0** | 2.68% |
| LastFM-2K | N@5 | 2.05 | 2.01 | 1.63 | 1.12 | 1.28 | 0.94 | 1.76 | 16.57 | 17.53 | **34.02** | 94.07% |
| | R@5 | 2.84 | 3.15 | 2.52 | 1.76 | 2.01 | 1.51 | 2.61 | 18.46 | 21.30 | **34.89** | 63.80% |
| | N@10 | 2.57 | 2.41 | 2.04 | 1.47 | 1.63 | 1.37 | 2.64 | 17.12 | 17.81 | **34.08** | 91.35% |
| | R@10 | 4.45 | 4.33 | 3.77 | 2.87 | 3.12 | 2.82 | 4.21 | 20.17 | 25.74 | **35.07** | 36.25% |
| ML-1M | N@5 | 3.46 | 3.65 | 3.46 | 1.65 | 25.75 | 21.29 | 8.04 | 75.62 | 2.30 | **100.0** | 32.24% |
| | R@5 | 5.55 | 5.83 | 5.55 | 2.52 | 28.14 | 26.32 | 10.76 | 83.19 | 3.76 | **100.0** | 20.21% |
| | N@10 | 4.55 | 4.88 | 4.65 | 2.25 | 26.34 | 22.29 | 9.39 | 76.97 | 3.17 | **100.0** | 29.91% |
| | R@10 | 9.27 | 9.67 | 9.29 | 4.40 | 29.95 | 29.39 | 14.96 | 87.37 | 5.75 | **100.0** | 14.46% |

Table 1: Performance comparison on four datasets. Results are reported as the mean over five independent runs. *Improv* denotes the relative improvement of FedRKG over the best performing baseline. S.K and D.K denote single knowledge and dual knowledge respectively.

## 6.2 Performance Comparison

Table 1 shows the recommendation performance on four real-world datasets. We summarize the results and discuss several key observations: **(1) FedRKG's Superiority over Centralized Methods.** FedRKG significantly outperforms centralized models (CenRec). Unlike CenRec's single global item embedding, FedRKG maintains personalized embeddings and injects global knowledge through guidance, allowing it to capture user-specific preferences effectively. **(2) Outperforming Single-Knowledge FedRec Models.** FedRKG surpasses other FedRecs that rely on a single knowledge source. These methods often suffer from personalization smoothing because they repeatedly replace local embeddings with global knowledge. In contrast, FedRKG preserves personal knowledge while selectively injecting global insights, achieving stronger performance. **(3) Surpassing Dual-Knowledge FedRec Models with Higher Efficiency.** Our method outperforms dual-knowledge models by adopting a more efficient strategy. Instead of learning and storing two separate embeddings, FedRKG treats global knowledge as complementary guidance. Consequently, FedRKG achieves the high performance of dual-knowledge systems with the memory footprint of single-knowledge models, making it a practical and powerful solution for on-device recommender. Taken together, these results align with the stepwise convergence in Fig. 4: each guidance round yields a discrete gain notably on FilmTrust and Amazon-Video.

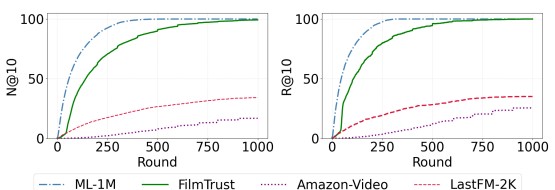

Figure 4: Convergence curves of FedRKG.

|  | Amazon-Video | | LastFM-2K | |
|---|---|---|---|---|
|  | N@5 | N@10 | N@5 | N@10 |
| FedRKG | **14.85** | **19.46** | **34.21** | **34.99** |
| w/o AG | 14.35 | 18.59 | 34.00 | 34.75 |
| w/ GIE | 2.44 | 3.72 | 1.43 | 2.29 |
| w/ PIE | 8.02 | 10.64 | 33.76 | 34.67 |

Table 2: Ablation on Amazon-Video and LastFM-2K. Variants: w/o AG = fixed weight guidance; w/ GIE = only global embeddings; w/ PIE = only local embeddings.

## 6.3 Analysis

**Ablation Study.** Table 2 reports ablations on two datasets with three variants: w/o AG (Knowledge Guidance with a fixed weight; no adaptive guidance), w/ GIE (global-only; full replacement by global item embeddings), and w/ PIE (personalized-only; no global guidance). We find three consistent trends: (1) Knowledge Guidance (KG) outperforms both global-only and personalized-only, showing that preserving personalization while fusing global trends is beneficial; (2) removing Adaptive Guidance degrades performance, indicating that per–user–item gating is better than a fixed coefficient; (3) even without adaptivity, the KG variant still beats other baselines, confirming the mechanism's effectiveness. Full results including R@5 and R@10 are provided in the Appendix.

|  | Amazon-Video | | FilmTrust | | LastFM-2K | | ML-1M | |
|---|---|---|---|---|---|---|---|---|
|  | N@10 | R@10 | N@10 | R@10 | N@10 | R@10 | N@10 | R@10 |
| FedMF | 3.04 | 5.61 | 26.56 | 33.82 | 1.79 | 3.39 | 4.65 | 9.34 |
| w/ AG | 16.80 | 25.51 | 99.22 | 100.0 | 34.23 | 35.07 | 100.0 | 100.0 |
| FedNCF | 1.16 | 2.62 | 20.78 | 28.20 | 1.95 | 3.70 | 2.39 | 4.65 |
| w/ AG | 5.27 | 8.97 | 97.18 | 99.10 | 26.28 | 28.84 | 99.91 | 100.0 |
| PFedRec | 3.78 | 5.69 | 35.34 | 41.73 | 1.80 | 3.62 | 25.78 | 29.24 |
| w/ AG | 16.28 | 24.78 | 97.97 | 99.51 | 31.70 | 33.81 | 99.73 | 100.0 |

Table 3: Performance comparison demonstrating the model-agnosticism of guidance mechanism.

**Model Agnosticism.** To examine whether our Adaptive Guidance can be applied to existing FedRec models, we integrate it into three distinct FedRec backbones: FedMF, FedNCF, and PFedRec. Table 3 shows that applying our paradigm brings substantial performance improvements. The gains are particularly significant on denser datasets like FilmTrust and ML-1M, where our method dramatically elevates the performance of all backbone models. Results validate that our paradigm is a general and highly effective approach for enhancing existing federated recommendation methods.

**Effect of Guidance**. Table 4 evaluates the effect of Knowledge Guidance (KG) and Adaptive Guidance (AG) on the Amazon-Video dataset. We first define popularity-based user groups on Amazon-Video: popular items are the top 30% by training interactions; users with a high share of interactions

| Method | Distinct Users | | | | Followers | | | |
|---|---|---|---|---|---|---|---|---|
| | N@5 | R@5 | N@10 | R@10 | N@5 | R@5 | N@10 | R@10 |
| w/ KG | -9.63% | -8.70% | -9.35% | -7.41% | 15.36% | 20.00% | 14.11% | 10.53% |
| | Improved users: 94.1%, Declined users: 5.9% | | | | | | | |
| w/ AG | **-4.54%** | **-4.35%** | **-2.52%** | **0.00%** | **29.51%** | **23.08%** | **23.15%** | **12.50%** |
| | Improved users: **94.68%**, Declined users: **5.32%** | | | | | | | |

Table 4: Guidance effect on Amazon-Video: relative change (%) per group (distinct vs. followers) under KG and AG. KG/AG denote Knowledge/Adaptive Guidance.

on these items are labeled *followers*, and those with a low share are labeled *distinct*. We sample 100 users from each group and measure the relative change immediately after a guidance round compared to just before it (at round 900); We first find that distinct users show small drops after guidance, implying that injecting global knowledge can hurt users whose preferences are highly personalized. Followers, in contrast, consistently improve. AG further mitigates the drop for distinct users (smaller drops than KG) and amplifies the gains for followers, indicating that AG injects less global signal for distinct users and more for followers. Finally, counting users whose aggregate score improved vs. declined shows that the vast majority improve, with a higher fraction under AG. These results confirm the effectiveness of the guidance mechanism, especially the adaptive variant.

**Performance on Cold vs. Warm Users.** Since our training inherently emphasizes personalization, users with fewer interactions (cold users) may perform worse compared to users with richer interaction histories (warm users). To verify whether our method can outperform dual-knowledge models specifically on cold users, we conduct an experiment where the top 30% of users (by interaction count) were categorized as warm and the bottom 30% as cold on the Amazon-Video dataset. Figure 5 shows that FedRKG achieves higher performance on cold users than FedRAP, which is the strongest dual-knowledge baseline on this dataset. This demonstrates that our model has stronger generalizability than models that explicitly maintain two types of knowledge.

Table 5: Performance under Local Differential Privacy on FilmTrust.

| | N@5 | R@5 | N@10 | R@10 |
|---|---|---|---|---|
| FedMF w/ DP | 18.79 | 21.19 | 20.90 | 27.87 |
| PFedRec w/ DP | 30.88 | 34.64 | 32.50 | 39.61 |
| FedRAP w/ DP | 89.89 | 92.67 | 90.44 | 94.38 |
| FedRKG w/ DP | **94.91** | **97.31** | **95.44** | **98.94** |

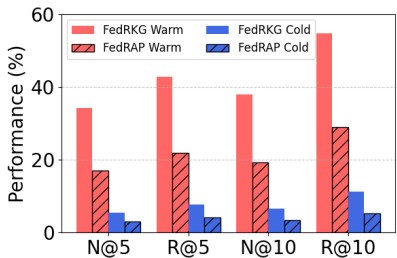

Figure 5: Warm vs. cold performance.

**FedRKG with Local Differential Privacy.** To examine whether FedRKG maintains strong performance under $(\epsilon, \delta)$-LDP, we conduct experiments with sensitivity set to 0.1 and $\sigma = 0.001$. We implement these experiments using the Opacus library (Yousefpour et al., 2021). The evaluation is performed on the FilmTrust dataset, where we additionally applied LDP to FedRAP, FedMF, and PFedRec. Table 5 shows that our proposed method consistently achieves high performance even in the presence of noise. Additional comparisons are provided in the Appendix.

## 7 CONCLUSION

In this work, we propose FedRKG, a novel paradigm that replaces the conventional full replacement strategy in federated recommendation with a guidance-based approach. FedRKG preserves personalization by maintaining local item embeddings as the primary knowledge source, while global knowledge is injected through a user–item level gating mechanism. This design enables FedRKG to achieve the memory efficiency of single-knowledge models while delivering the performance of dual-knowledge approaches. Furthermore, adaptive guidance and nested training allow personalized ratios of global knowledge injection, further boosting effectiveness. Extensive experiments across multiple datasets demonstrate that FedRKG consistently achieves state-of-the-art performance and can serve as a model-agnostic enhancement for existing federated recommendation frameworks.

## 8 REPRODUCIBILITY

We provide comprehensive details across the paper.

- **Source Code**: The complete sorce code, including instruction for setup and execution and data preprocessing code is available.
- **Hyerparameters and Training Details**: All hyperparameters, training configurations, and implementation details for our models and baselines are detailed in Appendix C.2.
- **Computational Environment**: The required computational environment and library dependencies are listed in the environment.yml file included with our source code.

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

# A  PROOF FOR PROPOSITION 1

*Proof.* Let $\mathbf{P}_g := \frac{1}{n} \sum_{u \in \mathcal{U}} \mathbf{P}_u$ be the *global knowledge* obtained by server-side aggregation (e.g., FedAvg). The objective is to minimize the squared L2 deviation between the personalized local item embeddings $\mathbf{P}$ and the global item embeddings $\mathbf{P}_g$:

$$\mathcal{L}(\mathbf{P}) = \tfrac{\lambda}{2} \|\mathbf{P} - \mathbf{P}_g\|_2^2, \quad \lambda > 0. \tag{15}$$

Its gradient is $\nabla_{\mathbf{P}} \mathcal{L} = \lambda(\mathbf{P} - \mathbf{P}_g)$. A single gradient-descent update with learning rate $\eta$ gives

$$\mathbf{P}' = \mathbf{P} - \eta \nabla_{\mathbf{P}} \mathcal{L} \tag{16}$$
$$= \mathbf{P} - \eta \lambda (\mathbf{P} - \mathbf{P}_g) \tag{17}$$
$$= (1 - \eta \lambda) \mathbf{P} + \eta \lambda \, \mathbf{P}_g. \tag{18}$$

Setting $\beta = 1 - \eta \lambda$ yields

$$\mathbf{P}' = \beta \, \mathbf{P} + (1 - \beta) \, \mathbf{P}_g, \tag{19}$$

which matches our Knowledge Guidance update and completes the proof.

Proposition 1 provides a key insight into our proposed mechanism. It demonstrates that Knowledge Guidance is not merely an ad-hoc heuristic for model interpolation. Instead, it can be interpreted as a global knowledge regularized update of the local model—a one-step gradient descent on a quadratic global knowledge alignment penalty. This single step moderately pulls the personalized item embeddings ($\mathbf{P}$) towards the generalized parameters of the global item embeddings ($\mathbf{P}_g$), encouraging the local model to learn from the global knowledge maintaining its unique specialization. This interpretation provides a clear theoretical justification for our approach in balancing the critical trade-off between personalization and generalization.

# B  FEDRKG'S OVERALL FLOW

FedRKG is trained using an alternating optimization algorithm, as detailed in Algorithm 1. The server first initializes the global item embeddings then distributes them to all users. Each user locally initializes private parameters $\mathbf{e}_u, \mathbf{M}_u, \mathbf{W}_u$ and $b_u$. In each communication round, the server randomly selects a subset of users to participate. These selected clients update their local parameters using their private data. If the round is not a designated guidance round, updated parameters

---

**Algorithm 1** FedRKG - Overall Process

---

**Input**: $\mathbf{R}, \eta, \eta_{\text{gate}}, T, T_{int}, E, E_{\text{gate}}$
**Initialize**: Global parameters $\mathbf{P}_g$

1: **for** each client $u \in \mathcal{U}$ **in parallel do**
2:     Initialize $\mathbf{P}_u, \leftarrow \mathbf{P}_g$ and $\mathbf{e}_u, \mathbf{M}_u, \mathbf{W}_u, b_u$
3: **end for**
4: **for** $t = 1, \ldots, T$ **do**
5:     $\mathcal{U}^t \leftarrow$ Randomly select $n_s$ clients
6:     **for** each $u \in \mathcal{U}^t$ **in parallel do**
7:         **LocalTrainRec**$(E, \eta)$
8:     **end for**
9:     **if** $t \% T_{int} == 0$ **then**
10:         — *Guidance* —
11:         Aggregate $\mathbf{P}_g$ from $\{\mathbf{P}_u\}_{u \in \mathcal{U}}$
12:         **for** each $u \in \mathcal{U}$ **in parallel do**
13:             **LocalTrainGate**$(\mathbf{P}_g, E_{\text{gate}}, \eta_{\text{gate}})$
14:         **end for**
15:         **for** each $u \in \mathcal{U}$ **in parallel do**
16:             Update $\mathbf{P}_u$ using $\mathbf{P}_g, \mathbf{W}_u, b_u$
17:         **end for**
18:     **end if**
19: **end for**

**LocalTrainRec**$(E, \eta)$:

1: **for** $e = 1, \ldots, E$ **do**
2:     Compute $\nabla \mathcal{L}_{rec}$ (Eq. 2) and update $\mathbf{P}_u, \mathbf{e}_u, \mathbf{M}_u$
3: **end for**

**LocalTrainGate**$(\mathbf{P}_g, E_{\text{gate}}, \eta_{\text{gate}})$:

1: **for** $e = 1, \ldots, E_{\text{gate}}$ **do**
2:     Compute $\nabla \mathcal{L}_{rec}$ (Eq. 7) and update $\mathbf{W}_u, b_u$
3: **end for**

---

remain local. If the round is a guidance round, the process includes additional steps. First, all clients upload their personalized item embeddings $\mathbf{P}_u$ to the server, which aggregates them to get global item embeddings $\mathbf{P}_g$. This updated $\mathbf{P}_g$ is then used in a nested training process where each client updates its local gating network parameters $\mathbf{W}_u, b_u$. Finally, each user update their personalized item embeddings using the newly updated gating network.

## C EXPERIMENTAL SETTINGS

### C.1 EXPERIMENTAL SETTINGS

| Dataset | #users | #items | #interactions | sparsity |
|---------|--------|--------|---------------|----------|
| Amazon-Video | 1,372 | 7,957 | 23,181 | 99.79% |
| FilmTrust | 1,002 | 2,042 | 33,372 | 98.37% |
| LastFM-2K | 1,269 | 12,322 | 183,415 | 98.83% |
| ML-1M | 6,040 | 3,706 | 1,000,209 | 95.53% |

Table C.1: Dataset statistics after preprocessing.

**Datasets.** We evaluate FedRKG on four real-world datasets: Amazon-Video[1] (Ni et al., 2019), FilmTrust[2] (Guo et al., 2013), LastFM-2K[3] (Cantador et al., 2011), and ML-1M[4] (Harper & Kon-

---

[1] https://jmcauley.ucsd.edu/data/amazon/
[2] https://guoguibing.github.io/librec/datasets
[3] https://grouplens.org/datasets/hetrec-2011/
[4] https://grouplens.org/datasets/movielens/

stan, 2015). These datasets have been widely adopted by prior federated recommendation works: Amazon-Video appears in (Zhang et al., 2023; He et al., 2024; Li et al., 2024; 2025), LastFM-2K in (Perifanis & Efraimidis, 2022; Zhang et al., 2023; 2024; He et al., 2024), FilmTrust in (He et al., 2024), and ML-1M in (Perifanis & Efraimidis, 2022; Zhang et al., 2023; 2024; He et al., 2024; Li et al., 2024; 2025). Leveraging this established suite allows us to evaluate FedRKG comprehensively across different domains, sizes, and sparsity levels. For data preprocessing, we filter out users with fewer than 20 interactions for the ML-1M dataset, and fewer than 10 interactions for the other three datasets. The detailed statistics of the datasets after preprocessing are summarized in Table C.1. All datasets are highly sparse, with a sparsity of over 95%, calculated as 1-(#Interactions/(#Users×#Items)). Although the datasets originally contain explicit ratings (from 1 to 5), we follow a widely adopted setting in recommendation research and convert them into implicit feedback(He et al., 2017; 2024; Li et al., 2024). Any record with a rating is treated as a positive sample. For each user, we split their interaction data based on timestamps. We sort their interactions chronologically and designate the most recent interaction for the test set, the second most recent for the validation set, and all remaining interactions for the training set.

**Evaluation.** We evaluate prediction performance using two widely used metrics: Recall (R@K) and Normalized Discounted Cumulative Gain (NDCG@K). Both criteria are defined in previous work (He et al., 2015). The @K denotes that for each user, we rank all unobserved items by their prediction scores and select the top K items. The Recall@K (R@K) is computed as the fraction of users for which the held-out test item appears in the top-K predicted items:

$$\text{R@K} = \frac{1}{|\mathcal{U}|} \sum_{u \in \mathcal{U}} \mathbb{I}(i_u^{\text{test}} \in \text{TopK}_u), \tag{20}$$

where $i_u^{\text{test}}$ denotes the test item for user $u$', and $\text{TopK}_u$ refers to the top $K$ items with the highest predicted scores among the unobserved items for user $u$. The Normalized Discounted Cumulative Gain (NDCG@K) additionally considers the rank of the relevant item:

$$\text{NDCG@K} = \frac{1}{|\mathcal{U}|} \sum_{u \in \mathcal{U}} \frac{1}{\log_2(r_u + 1)}, \tag{21}$$

where $r_u$ is the ranking (from 1) of the ground-truth item in the top-K list (or 0 if not present). R@K then measures whether the ground-truth test item is present in this top-K list, while NDCG@K additionally considers the ranking quality, assigning a higher score if the relevant item is ranked closer to the top. In this study, we set K to 5 and 10. While prior works (e.g., (Li et al., 2024; Zhang et al., 2024; 2023)) performed fast evaluation by ranking the test item against 99 randomly sampled negatives, we, for fairness, rank the test item against the entire set of unobserved items for each user and report metrics from this full-ranking evaluation.

**Baselines.** We compare the performance of FedRKG against several centralized methods and state-of-the-art federated settings.

- **MF** (Koren et al., 2009): A classic recommendation algorithm that factorizes the rating matrix into user and item latent embeddings. The prediction score is computed via an inner product.

- **NeuMF** (He et al., 2017): A popular model that replaces the inner product of MF with a multi-layer perceptron (MLP) to learn the complex user-item interaction function.

- **FedMF** (Chai et al., 2020): A federated version of MF where user embeddings are trained locally while item embeddings are aggregated on the server.

- **FedNCF** (Perifanis & Efraimidis, 2022): The federated version of NeuMF. User embeddings are updated locally, while the item embeddings and MLP layers are aggregated.

- **PFedRec** (Zhang et al., 2023): A personalized FedRec method that learns a local scoring function for each user and employs a two-step optimization process.

- **GPFedRec** (Zhang et al., 2024): A personalized FedRec model that constructs a user-relation graph on the server based on embedding similarity to generate optimized embeddings for each user.

- **CoFedRec** (He et al., 2024): A personalized FedRec model that introduces co-clustering by first clustering items and then dynamically forming user groups based on their preferences for these item clusters.

- **FedRAP** (Li et al., 2024): A pioneering dual-knowledge model where each user learns both a local (personalized) and a global (shared) item embedding.
- **FedDAE** (Li et al., 2025): A dual-knowledge model that uses a variational autoencoder for non-linearity and a gating network for the adaptive combination of personalized and shared knowledge.

### C.2    IMPLEMENTATION DETAILS

All training and inference are conducted using PyTorch with CUDA on an RTX 3090 GPU and AMD EPYC 7313 CPU. Following previous works (He et al., 2017; Zhang et al., 2023; Li et al., 2024), we randomly select four negative samples for each positive sample in the training set. To ensure a fair comparison, we fix the latent embedding dimension to 32 and the training batch size to 256 for all methods. We use PyTorch (Paszke et al., 2019) and the Stochastic Gradient Descent (SGD) optimizer for all parameter updates. We select learning rate and the gate learning rate from $\{10^{-4}, 10^{-3}, 10^{-2}, 10^{-1}\}$ for the FedRKG and all baselines. We apply early stopping with a patience of 100 epochs. In our main experiments, we use FedMF as the backbone model, where $\mathbf{M}_u$ is treated as an identity function. Under this setting, the prediction score is computed as $\hat{r}_{ui} = \mathbf{e}_u^\top \mathbf{P}_{ui}$. We set the number of local epochs for the main task and the nested gate training to 1 and 5, respectively. For all baselines and FedRKG to be converged, the maximum number of communication rounds is set to 3,000 for LastFM-2K and 1,000 for the other datasets. To obtain fine-grained, per-round trajectories (and to avoid the smoothing/confounding effect of larger local epochs with guidance every round), we deliberately fix the local epoch to $E{=}1$ and schedule guidance at fixed intervals rather than every round. The guidance is applied every 100 communication rounds. We set $n_s$ to $n$. $\beta$ is set to 0.999 for LastFM-2K and 0.99 for the other datasets. We set $\alpha_u = 1/n$ for the standard federated averaging scheme for all federated baselines. No pre-training strategies or additional privacy-preserving measures are applied in our main experiments, ablation study and hyperparameter study.

Each experiment is repeated five times with different seeds. For every run we fix all random-number generators in Python, NumPy, and PyTorch to the chosen seed, synchronise the CUDA RNG across devices, set PYTHONHASHSEED accordingly, disable the CuDNN autotuner (*benchmark*=false), and enforce deterministic kernels. Data-loader workers are reseeded with *seed + worker_id*. Sensitivity-analysis and convergence plots show the trajectory obtained with seed 42.

For baselines, we set the embedding (or latent) dimension to 32 for all models. Model-specific hyper-parameters are searched as follows.

- **MF**: We use binary cross-entropy loss for training, although original MF designed for explicit feedback using rating-based regression (e.g., MSE loss).
- **NeuMF**: We use both a GMF and an MLP component. The MLP consists of 3 layers $(64 \to 32 \to 16 \to 8)$, and the final prediction is computed by concatenating the outputs of the GMF and MLP branches.
- **FedMF**: We use a federated version of matrix factorization (MF) without encryption.
- **FedNCF**: We employ a 2-layer MLP for the scoring function with architecture $64 \to 32 \to 1$.
- **PFedRec**: We use a personalized 2-layer MLP for scoring, with dimensions $32 \to 16 \to 1$.
- **GPFedRec**: We adopt a 4-layer MLP architecture $(64 \to 32 \to 16 \to 8 \to 1)$. We set the neighborhood similarity threshold to 0.5 and search the regularization coefficient over the same space as the learning rate.
- **CoFedRec**: We follow the original paper's configuration. For the Amazon-Video dataset which is not used in the original paper, we set the contrastive temperature to $\tau = 0.5$, the regularization coefficient to $10^{-3}$, and the number of item clusters to 30.
- **FedRAP**: We use two regularization coefficients $v_1$ and $v_2$, modulated by a $\tanh$ annealing schedule as described in the original paper.
- **FedDAE**: We set the latent dimension to 64 (32 for mean, 32 for variance). We use a 2-layer encoder $(m \to 64 \to 64)$ and a 2-layer decoder $(32 \to 64 \to m)$, and adopt the official implementation provided by the authors.

## D  FURTHER ANALYSIS

|  | Amazon-Video | | | | LastFM-2K | | | |
|---|---|---|---|---|---|---|---|---|
|  | N@5 | R@5 | N@10 | R@10 | N@5 | R@5 | N@10 | R@10 |
| FedRKG | **14.85** | **19.46** | **16.80** | **25.51** | **34.21** | **34.99** | **34.23** | **35.07** |
| w/o AG | 14.35 | 18.59 | 16.37 | 24.93 | 34.00 | 34.75 | 34.11 | **35.07** |
| w/ GIE | 2.44 | 3.72 | 3.04 | 5.61 | 1.43 | 2.29 | 1.79 | 3.39 |
| w/ PIE | 8.02 | 10.64 | 9.27 | 14.50 | 33.76 | 34.67 | 33.90 | **35.07** |

Table D.1: Full ablation on Amazon-Video and LastFM-2K.

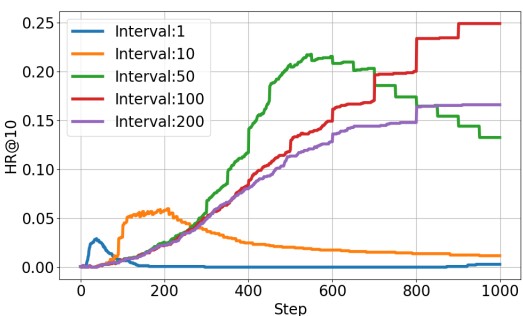

Figure D.1: Effect of the guidance interval on convergence (HR@10 vs. rounds, Amazon-Video). Global knowledge is injected every $T_{int}$ rounds with $T_{int} \in \{1, 10, 50, 100, 200\}$.

**Interval Analysis.** We further observe how convergence curves are shaped under different guidance intervals when applying Knowledge Guidance, as shown in Figure D.1. This figure corresponds to the setting without Adaptive Guidance. When the guidance interval is too small, global knowledge is injected repeatedly before sufficient personalized knowledge can be formed, leading to poor performance. As the interval increases, performance improves, since users' personalized knowledge has more time to develop. However, if the interval becomes too large, the model is updated almost exclusively through personalized interactions, causing a lack of global knowledge. In this case, convergence slows and performance only rises sharply when global knowledge is eventually injected. We also find that the curves typically peak and then decline. This suggests that once sufficient knowledge has been established, user-specific uniqueness should be preserved. However, the aggregated global knowledge—formed by combining these diverse personalized representations—can behave like noise, thereby harming performance.

Table D.2: Performance comparison with/without Local Differential Privacy on FilmTrust dataset

|  | N@5 | R@5 | N@10 | R@10 |
|---|---|---|---|---|
| FedMF w/o DP | 24.99 | 27.22 | 26.56 | 33.82 |
| FedMF w/ DP | 18.79 | 21.19 | 20.90 | 27.87 |
| PFedRec w/o DP | 33.92 | 37.33 | 35.34 | 41.73 |
| PFedRec w/ DP | 30.88 | 34.64 | 32.50 | 39.61 |
| FedRAP w/o DP | 92.35 | 95.93 | 92.76 | 97.15 |
| FedRAP w/ DP | 89.89 | 92.67 | 90.44 | 94.38 |
| FedRKG w/o DP | 99.11 | 99.67 | 99.22 | 100.0 |
| FedRKG w/ DP | **94.91** | **97.31** | **95.44** | **98.94** |

**Ablation Study on Local Differential Privacy.** We additionally evaluate LDP-enhanced versions of the federated methods and compare them with their original counterparts in Table 1 and Table D.2. The results indicate that FedRKG under $(\epsilon, \delta)$-LDP retains its performance advantage while also providing privacy guarantees.

**Analysis of Gating for Distinct Users vs. Followers.** Using the same popularity-based grouping defined in the Section 6.3, Table D.3 compares gating values (logits) on popular items between distinct users and followers. For each group, we average gating values over the popular item set and report the relative (%) change between followers and distinct users. For each group, we form a group mean over the popular-item set: for the gating *value* we average $g_{u,i}$, and for the gating *logit*

|  | w/ 100 users | w/ 300 users | w/ 500 users |
|---|---|---|---|
| gating value | 0.1% | 0.04% | 0.06% |
| gating logit | 34.55% | 15.79% | 20.69% |

Table D.3: Relative change (%) in gating *value/logit* from distinct users to followers. $\Delta(\%) = 100 \times \frac{g_F - g_D}{g_D}$, where $g$ is the group mean over popular items (*value*: average $g_{u,i}$; *logit*: average $\sigma^{-1}(g_{u,i})$).

we average $\sigma^{-1}(g_{u,i})$. We then report the relative change from distinct to followers as

$$\Delta(\%) = 100 \times \frac{g_F - g_D}{g_D},$$

where $g_F$ and $g_D$ denote the corresponding group means for followers and distinct users, respectively. Results show that distinct users have consistently lower gates on popular items, indicating that the learned gate tracks behavior: followers accept more global knowledge for popular items, whereas distinct users rely on it less.

|  | 10% Users | 20% Users | 30% Users |
|---|---|---|---|
| gating value | 0.32% | 0.26% | 0.22% |
| gating logit | 288% | 163% | 110% |

Table D.4: Relative change (%) in gating *value/logit* from cold to warm users. $\Delta(\%) = 100 \times \frac{g_W - g_C}{g_C}$, where $g$ is the group mean over all items (*value*: average $g_{u,i}$; *logit*: average $\sigma^{-1}(g_{u,i})$). Warm = top-$k$% and Cold = bottom-$k$% by interaction count.

**Analysis of Gating for Cold vs. Warm Users.** Using the interaction–count grouping from Section 6.3, we compare gating on all items between warm (top-$k$% users by interactions) and cold (bottom-$k$% users) users on Amazon-Video and FilmTrust. For each group, we form a group mean over the popular-item set: for the gating *value* we average $g_{u,i}$, and for the gating *logit* we average $\sigma^{-1}(g_{u,i})$. We then report the relative change from cold to warm as

$$\Delta(\%) = 100 \times \frac{g_W - g_C}{g_C},$$

where $g_W$ and $g_C$ denote the corresponding group means for warm and cold users, respectively (for the logit metric, $g$ uses $\sigma^{-1}(g_{u,i})$). Table D.4 shows that cold users exhibit higher gates on all items than warm users, suggesting that when personal training data is scarce, users rely more on global knowledge.

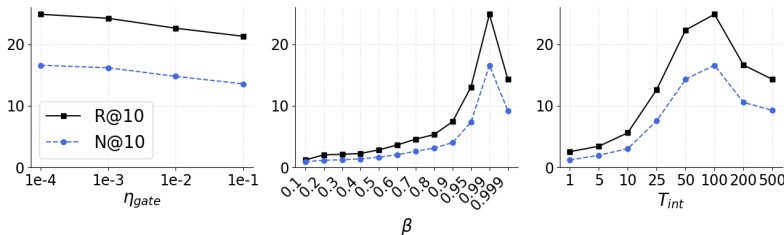

Figure D.2: Sensitivity analysis on Amazon-Video for gate learning rate $\eta_{\text{gate}}$, retention $\beta$, and guidance interval $T_{\text{int}}$ (rounds), measured by N@10/R@10.

**Sensitivity Analysis.** Figure D.2 shows the sensitivity analysis of key hyperparameters: the learning rate for the gating network $\eta_{gate}$, the momentum parameter $\beta$, and the guidance interval $T_{int}$. We conduct the analysis on the Amazon-Video dataset and report N@10 and R@10, noting that similar tendencies are observed on other datasets. First, we observe that performance decreases as $\eta_{gate}$ increases. A higher gate learning rate appears to prevent the model from learning the guidance ratio effectively, leading to degraded performance. For $\beta$, we see that performance improves as the value increases up to 0.99, but then drops at 0.999. This indicates that while personalized knowledge is crucial, simply maximizing its retention is suboptimal; a proper amount of guidance from global

knowledge is necessary for the best results. Similarly, the performance improves as the guidance interval increases to 100, but declines thereafter. This suggests that guidance is ineffective if applied too frequently or too infrequently, highlighting the importance of finding an optimal balance.

## E    DISCUSSION AND LIMITATIONS

While Knowledge Guidance yields step-wise gains with a single-embedding footprint, several limitations remain. The effect depends on the quality of the global knowledge and the guidance schedule; extremely non-stationary environments or poorly aggregated parameters can dilute gains. A small subset of highly distinctive users can experience immediate dips after guidance; our adaptive variant reduces but does not eliminate this behavior. As future work, we aim to make guidance more robust to the quality global knowledge and to mitigate post-guidance drops—e.g., by confidence-weighted global injection, per-user scheduling, and robust aggregation of global knowledge.

## F    LARGE LANGUAGE MODEL USAGE

We use large language models solely for grammar checking and minor wording polish. All research ideas, methodology, analyses, and conclusions are original to the authors.

