# OpenReview forum: "Personalized Federated Recommendation with Knowledge Guidance"
_ICLR.cc/2026/Conference — ICLR 2026 Conference Withdrawn Submission_

### Official Review · Reviewer_9NJ7 · 2025-10-27

**Soundness:** 1
**Presentation:** 2
**Contribution:** 1
**Rating:** 2
**Confidence:** 5

**Summary:**

The paper proposes FedRKG, a lightweight framework that improves personalization in federated recommendation systems without increasing memory cost. Instead of fully replacing local knowledge with global updates, FedRKG introduces knowledge guidance, which fuses global trends into local embeddings. This approach achieves the accuracy of dual-knowledge models with the efficiency of single-knowledge systems, showing superior performance across multiple benchmark datasets while maintaining strong privacy through local differential privacy protection

**Strengths:**

- The paper is easy to follow.
- The experiments are solid and extensive.

**Weaknesses:**

The claimed novelty of this paper is rather limited.

First, the authors repeatedly emphasize the notions of single-knowledge and dual-knowledge models (e.g., Line 13 and Line 41). However, these concepts directly correspond to the well-established global model and personalized model formulations in Personalized Federated Learning (PFL). The paper simply redefines existing PFL terminology without introducing any conceptual advancement.

Second, the proposed Knowledge Guidance mechanism is essentially equivalent to the classic regularization-based optimization used in PFL (e.g., Ditto, FedProx, pFedMe). It lacks theoretical or methodological novelty and offers no new insight beyond existing work.

Third, the paper motivates its method by claiming improved memory efficiency. However, in collaborative filtering scenarios, the model’s embedding parameters are already lightweight. The authors should have quantitatively compared the memory footprint of the proposed “dual embedding” setup versus the local dataset size. For typical edge devices, this additional memory cost is negligible, undermining the central argument of the paper.

Moreover, the Adaptive Guidance module, which modulates the fusion of local and global knowledge, appears abruptly in the method section. The motivation for introducing this component to balance item distribution bias between global and local embeddings is not clearly stated or theoretically supported in earlier sections. This inconsistency weakens the logical flow and the claimed contribution.

Finally, the paper asserts that single/dual knowledge structures are suboptimal and that FedRKG provides a better optimization scheme. However, this is largely unrelated to the recommendation domain itself and merely reproduces existing findings from the PFL literature. Numerous PFL studies over the past five years have already investigated the trade-off between local and global optimization bias and provided solid theoretical foundations. The current paper simply adopts these ideas without introducing new insights, formulations, or analyses.

**Questions:**

Please refer to the weaknesses.

---

### Official Review · Reviewer_q5KV · 2025-10-28

**Soundness:** 2
**Presentation:** 3
**Contribution:** 2
**Rating:** 2
**Confidence:** 4

**Summary:**

This paper proposes a personalized federated recommendation framework called FedRKG (Federated Recommendation with Knowledge Guidance), which aims to resolve the trade-off between the insufficient personalization of single-knowledge models and the excessive memory overhead of dual-knowledge models. The core idea is to replace “knowledge substitution” with “knowledge guidance.” To achieve this, the authors design an adaptive gating mechanism that dynamically controls the strength of global knowledge injection at the user–item level, thereby enhancing personalized recommendation performance. Experiments on multiple benchmark datasets demonstrate that the proposed method achieves significant performance improvements.

**Strengths:**

1. The gating network learns the optimal knowledge fusion strength for each user–item pair, enabling the model to dynamically adapt to user heterogeneity and improve recommendation accuracy.

2. The proposed mechanism can be seamlessly integrated into various federated recommendation frameworks (e.g., FedMF, FedNCF, PFedRec), exhibiting strong model-agnosticism and scalability.

3. Experimental results show that the method consistently improves performance across multiple datasets and outperforms existing baseline models.

**Weaknesses:**

1. Lack of novelty: The proposed concept of “knowledge guidance” essentially fuses the global and local models, which highly overlaps with existing personalized federated learning approaches such as FedALA[1]. The conceptual innovation is therefore limited. Moreover, the notion that directly substituting the local model with the global one is suboptimal in federated settings is already well known; model merging has long been recognized as a standard solution to address client heterogeneity. The paper also fails to cite or compare with relevant works such as FedALA.

2. Superficial theoretical analysis: The theoretical discussion merely shows that the update process is equivalent to adding a proximal regularization term on global parameters (similar to FedProx[2] or pFedMe[3]). However, it does not explain why the proposed mechanism outperforms standard regularization approaches, nor does it provide convergence or optimization analysis. Comparisons with FedProx and pFedMe are also missing.

3. Questionable experimental results: The reported performance far exceeds that of centralized models by even an order of magnitude, which is logically implausible (For ML-1M, MF: 3.46, FedMF: 3.46, Yours:100.00). This raises concerns about potential errors in evaluation procedures or metric computation. The paper provides no reasonable explanation for these anomalies.

4. Efficiency and practicality concerns: Although the paper claims that the introduction of the gating network and periodic guidance is efficient, it does not quantify the corresponding communication or computational overhead. The privacy protection component only superficially applies local differential privacy (LDP) with negligible noise, offering limited protection. Furthermore, all experiments are conducted on small-scale datasets, lacking validation on large-scale or real-device scenarios, which undermines the practical significance of the work.

Reference:

[1] Zhang, Jianqing, et al. "Fedala: Adaptive local aggregation for personalized federated learning." AAAI, 2023.

[2] Li, Tian, et al. "Federated optimization in heterogeneous networks." MLSys, 2020.

[3] T Dinh, Canh, Nguyen Tran, and Josh Nguyen. "Personalized federated learning with moreau envelopes." NeurIPS, 2020.

**Questions:**

1. Figure 2 (Preliminary experiments) presents information with high density, yet the paper provides insufficient explanation of key implementation details such as parameter settings and experimental configurations.

2. Although the paper emphasizes the differences among several local paradigms, its justification for the effectiveness of the proposed knowledge guidance mechanism over other paradigms remains purely intuitive, lacking theoretical or empirical analysis to substantiate the claimed advantages.

3. A substantial portion of related work in federated learning is missing from comparison, including representative methods such as FedALA, FedProx, and pFedMe.

---

### Official Review · Reviewer_9pxP · 2025-10-31

**Soundness:** 2
**Presentation:** 2
**Contribution:** 2
**Rating:** 4
**Confidence:** 5

**Summary:**

This paper introduces FedRKG, which mitigates the memory–personalization dilemma in local recommenders by leveraging globally aggregated item embeddings. Extensive experiments validate its superiority in top-k recommendation, ablation, compatibility, and privacy-preserving analyses.

**Strengths:**

1. The paper diagnoses the dilemma between suboptimal knowledge replacement practice in single-knowledge (only local item embeddings) and doubling memory in dual-knowledge models (local and global item embeddings).

2. A theoretical account of knowledge guidance is provided.

**Weaknesses:**

1. In section 4.3, Eq. (5) inputs to the user-specific gating network are dimensionally incompatible.

2. Experimental Limitations：

(1) Datasets vary across experiments. (Amazon-Video for cold/warm users, FilmTrust for LDP).
(2) Incomplete ablation (only Amazon-Video and LastFM-2K).
(3) Inconsistent FedRKG scores between Table 1 and Table 3.

**Questions:**

1. How does your method differ from and advance beyond [1]?
2. Why does FedRKG achieve 100 % accuracy on certain datasets?
3. In the ablation study, w/o AG performs almost on par with FedRKG, leaving the efficacy of adaptive guidance insufficiently demonstrated.
4. The hyper-parameter analysis should examine the number of nested gate training, and guidance interval across all datasets.

[1]	Chen J, Zhang H, Zhang C, et al. Breaking the Aggregation Bottleneck in Federated Recommendation: A Personalized Model Merging Approach[J]. arXiv preprint arXiv:2508.12386, 2025.

---

### Official Review · Reviewer_eUEm · 2025-10-31

**Soundness:** 2
**Presentation:** 2
**Contribution:** 2
**Rating:** 2
**Confidence:** 4

**Summary:**

This paper focuses on addressing the balance issue between memory efficiency and personalization performance for single-knowledge-based and dual-knowledge-based models in Federated Recommendation systems. It proposes the Knowledge-Guided Federated Recommendation framework, which enables the fusion of global knowledge into preserved local embeddings and incorporates adaptive guidance to dynamically adjust the intensity of global knowledge.

**Strengths:**

1. The paper is well-structured and written fluently, with clear logic in introducing the research background, problem formulation, framework design, and experimental validation.
2. The proposed FedRKG framework is simple and effective: it achieves dual-knowledge-level personalization while maintaining a single-knowledge memory footprint through KG, avoiding the memory overload of dual-knowledge models, which aligns with the practical demand for on-device FedRec deployment.
3. The framework exhibits good compatibility.

**Weaknesses:**

1. Inconsistencies exist between the performance results of FedRKG in Table 1 and Table 2. For example, in the Amazon-Video dataset: Table 1 reports FedRKG’s N@5=15.74 and N@10=17.50, while Table 2 (ablation study) shows FedRKG’s N@5=14.85 and N@10=19.46; The paper provides no explanation for this discrepancy.
2. The contribution of AG is contradictory across different experiments, and the paper lacks in-depth analysis of the degree of its contribution and applicable scenarios. In the ablation study (Table 2), removing AG (w/o AG) only causes a slight performance drop: on the Amazon-Video dataset, N@10 decreases from 19.46 to 18.59; however, in the model-agnosticism experiment (Table 3), adding AG to the backbone models leads to a significant performance improvement: the N@10 of FedMF on the Amazon-Video dataset jumps from 3.04 to 16.80. The paper fails to analyze why the contribution of AG varies drastically under different experimental settings, nor does it clarify under which backbone models AG brings more significant improvements and under which ones its effect is weaker.
3. The core mechanism KG lacks sufficient discussion on its novelty compared with existing studies, and the paper inadequately reviews related literature. The paper does not clearly elaborate on the differences between KG and these existing weighted fusion strategies, nor does it highlight its unique contributions, leading to ambiguous innovation of the core idea.

**Questions:**

Please refer to the weaknesses.

---

### Note · Authors · 2025-11-13

I have read and agree with the venue's withdrawal policy on behalf of myself and my co-authors.